# Deposition of Sol–Gel ZnO:Mg Films and Investigation of Their Structural and Optical Properties

**DOI:** 10.3390/ma15248883

**Published:** 2022-12-12

**Authors:** Tatyana Ivanova, Antoaneta Harizanova, Tatyana Koutzarova, Benedicte Vertruyen, Raphael Closset

**Affiliations:** 1Central Laboratory of Solar Energy and New Energy Sources, Bulgarian Academy of Sciences, Tzarigradsko Chaussee 72, 1784 Sofia, Bulgaria; 2Institute of Electronics, Bulgarian Academy of Sciences, Tzarigradsko Chaussee 72, 1784 Sofia, Bulgaria; 3GREENMAT, Institute of Chemistry B6, University of Liege, B6a Quartier Agora, Allee du Six Août, 13, 4000 Liège, Belgium

**Keywords:** sol–gel technology, Mg-doped ZnO films, optical band gap tuning, film morphology

## Abstract

This work presents a facile sol–gel method for the deposition of ZnO and ZnO:Mg films. The films are spin coated on silicon and quartz substrates. The impact of magnesium concentrations (0, 0.5, 1, 2 and 3 wt%) and post-annealing treatments (300–600 °C) on the film’s structural, vibrational and optical properties is investigated. Undoped ZnO films crystallize in the wurtzite phase, with crystallite sizes ranging from 9.1 nm (300 °C) to 29.7 nm (600 °C). Mg doping deteriorates the film crystallization and shifting of 002 peak towards higher diffraction angles is observed, indicating the successful incorporation of Mg into the ZnO matrix. ZnO:Mg films (2 wt%) possess the smallest crystallite size, ranging from 6.2 nm (300 °C) to 25.2 nm (600 °C). The highest Mg concentration (3 wt%) results into a segregation of the MgO phase. Lattice constants, texture coefficients and Zn–O bond lengths are discussed. The diminution of the *c* lattice parameter is related to the replacement of Zn^2+^ by Mg^2+^ in the ZnO host lattice. The vibrational properties are studied by Fourier transform infrared (FTIR) spectroscopy. IR lines related to Mg–O bonds are found for ZnO:Mg films with dopant concentrations of 2 and 3 wt%. The optical characterization showed that the transmittance of ZnO:Mg thin films increased from 74.5% (undoped ZnO) to about 89.1% and the optical band gap energy from 3.24 to 3.56 eV. Mg doping leads to a higher refractive index compared to undoped ZnO films. The FESEM (field emission scanning electron microscopy) technique is used for observation of the surface morphology modification of ZnO:Mg films. The doped ZnO films possess a smoother grained surface structure, opposite to the wrinkle-type morphology of undoped sol–gel ZnO films. The smoother surface leads to improved transparency of ZnO:Mg films.

## 1. Introduction

ZnO is a wide bandgap oxide semiconductor with a direct energy gap of about 3.37 eV, a high excitonic binding energy (60 meV) and high optical transparency in the visible and near-infrared ranges [1,2]. Other attractive properties of ZnO include its controllable electron mobility, low manufacturing cost, environmentally friendly nature, non-toxicity and easy availability [3]. Undoped ZnO films usually contain native defects such as zinc interstitials (Z_ni_) and oxygen vacancies (V_O_), and due to them, undoped ZnO is generally an intrinsic n-type semiconductor [4].

All these properties allow ZnO materials to be applied in different fields, such as UV optoelectronic devices, laser diodes, solar cells, sensors, field effect transistors, light emitting devices, photoconductive ultraviolet detectors, anti-reflective coatings and photocatalysis [3,5]. ZnO has enormous potential for optoelectronic applications. Doping with different metal and non-metal elements is a well-known approach for extending the advantages of ZnO by improving its structural, electrical, optical and physical properties [6]. Doped ZnO materials expand their applicability and multifunctionality [7]. Therefore, it is essential to dope ZnO with suitable elements and appropriate doping concentrations in order to obtain the desired enhanced optical and electrical behavior [7,8] or to improve specific properties, such as magnetic properties, photocatalytic performance, etc. [9]. It has been reported that the enhancement of the electrical and optical properties of n-type ZnO material can be achieved by doping with Al, In, Ga, Sn and Si, V, Nb [10]. For p-type conductive ZnO, various dopants, including chemical elements from group V (N, P, As, Sb), group I (Li, Na, K), and group IB (Ag, Cu), have been suggested as acceptors [11]. The photocatalytic activity of ZnO can be stimulated with a particular amount of rare earth elements, such as La, Eu, Ce and Sm [12], or with metal dopants, such as Co, Ni, Mn, Bi, Cu, Ag, etc. [13]. Effective transition-metal-doped ZnO has been proven to be the most promising room-temperature ferromagnetic material [14]. In particular, Co-doped ZnO dilute magnetic semiconductors (DMS) have raised a large interest in recent decades, because no phase transition occurs when the molar amount of cobalt dopant reaches 0.146 [15]. Moreover, doping with Fe element increases the electrical and dielectric properties of ZnO, which renders ZnO:Fe useful for electric storage applications [16]. Co-doping, namely, doping two or more kinds of extrinsic atoms simultaneously, is considered as an effective method for improving the properties of ZnO [17]. For example, Fe/Al co-doping in ZnO induces better optical, structural and electrical characteristics and makes this material useful for optoelectronic applications [18]. Modification of the ZnO surface can improve the properties of bulk heterojunction (BHJ) structure-based organic photovoltaics [19].

Recently, it has been reported that Mg doping in ZnO has considerable influence on its optical properties [20]. Mg addition leads to band gap engineering and to a decrease in the oxygen vacancies and electron density of ZnO materials [20,21]. It is known that the ionic radii of Mg^2+^ (0.57 Å) and Zn^2+^ (0.60 Å) are comparable and the incorporation of Mg^2+^ in the ZnO lattice can produce a compound material with less lattice distortions [22]. Mg-doped ZnO thin films are investigated due to their room-temperature ferromagnetic properties [23], piezoelectric quality [24], high transmittance and wide band gap [25], and sensing characteristics [26]. Mg_x_Zn_1−x_O transparent semiconductor thin films find applications in visible-blind, ultraviolet-sensitive photoconductive detectors [27] and as piezoelectric materials [28].

ZnO and doped ZnO films can be deposited by various techniques, including chemical vapor deposition, atomic layer deposition, pulsed laser deposition, thermal evaporation, radio-frequency sputtering, and molecular beam epitaxy [29,30]. Among these deposition methods, the sol–gel technique has major advantages, including its safety, low cost, compositional control and large surface area coating capability, relatively low crystallization temperature and uniform film thickness, and lack of need for complex and expensive vacuum equipment, [31,32].

Successful deposition of Mg-doped ZnO thin films has previously been reported using the sol–gel method [33,34]. ZnO:Mg films can be spin coated [34] or dip coated [35]. The increase in the optical band gap to 3.36 eV with increasing Mg concentration was reported [34], confirming that Mg ions were incorporated into the ZnO host lattice [34]. Sol–gel prepared thin films of ZnO:Mg with tuned characteristics would have promising applications in optoelectronic devices [36].

This study is aimed at the synthesis and the structural, optical, electrical and morphological characterization of ZnO:Mg thin films deposited by a simple and reproducible sol–gel route with four dopant concentrations (0.5, 1, 2 and 3 wt%), as a function of annealing temperature (300–600 °C). Undoped ZnO films are prepared for comparative analysis. The influence of Mg doping on the structural, vibrational, optical and morphological properties of the sol–gel derived thin films are systematically investigated, and the obtained results are discussed. ZnO:Mg films achieve a high transparency, smooth surface and nanocrystalline structure. The sheet resistance of undoped ZnO films is decreased from 688 to 240 Ω/sq with Mg incorporation. The optimal Mg doping concentration in ZnO films is found regarding structural, optical and electrical properties. The proposed facile sol–gel spin coating method is suitable for the fabrication of high quality and nanostructured Mg-doped ZnO films on quartz and silicon substrates.

## 2. Materials and Methods

Sol solution of 0.4 M concentration for ZnO film deposition was synthesized according to a previously reported procedure [37]. The precursor was zinc acetate dihydrate (Zn(CH_3_COO)_2_.2H_2_O, Riedel de Haen, Hannover, Germany), the solvent was absolute ethanol (C_2_H_5_(OH), Merck KgaA Darmstadt, Germany, absolute for analysis) and the complexing agent was monoethanolamine (MEA, Fluka AG, Buchs, Switzerland, 98%) with an MEA/Zn molar ratio equal to 1 [37]. The obtained sol was transparent and without precipitation. The sol solution for ZnO deposition was found to be stable and exhibited good film forming properties for 4 months. Mixed sols for the preparation of Mg-doped ZnO films were prepared by dissolving 0.5, 1, 2 and 3 wt% of magnesium acetate (Mg(CH_3_COO)_2_.4H_2_O, Merck, Darmstadt, Germany) into an appropriate ZnO sol volume. The mixed sols for ZnO:Mg formation were stirred using a magnetic stirrer (ARE, Velp Scientifica s.r.l., Usmate, Italy) at 50 °C/2 h, followed by treatment in an ultrasonic bath (ultrasonic cleaner, UST 2.8-100, Siel Ltd., Gabrovo, Bulgaria) at 45 °C/2 h. The final sol solutions were homogeneous and limpid. The mixed sols and the corresponding thin films are labelled as ZnO:Mg 0.5, ZnO:Mg 1. ZnO:Mg 2 and ZnO:Mg 3.

Undoped ZnO and ZnO:Mg thin films were deposited by spin coating (Spin coater P 6708, PI-KEM Limited, Staffordshire, UK) on preliminary cleaned Si (p-type, orientation <100>, resistivity 4.5–7.5 Ω) and quartz (UV graded, glass thickness 1 mm ± 0.1) substrates. Sol solution was dropped on the substrates, followed by rotating the substrates at 2000 rpm/30 s. After the spin coating process, the films were dried at 300 °C for 10 min in a furnace in order to evaporate the solvent and to remove organic residuals (preheating procedure). The coating and preheating procedure was reiterated five times to fabricate thicker films. Then, the obtained films were subjected to a high annealing temperature of 300, 400, 500 and 600 °C for 1 h in air ambient with a controlled constant heating and cooling rate of 10 °C/minute.

Si wafers (FZ, p-type, resistivity 4.5–7.5 Ω, orientation <100>, Wacker-Chemitronic GMBH, Munich, Germany) were applied for structural and surface morphology research and UV graded quartz glass substrates (thickness 1 mm ± 0.1, Präzisions Glas & Optik GmbH, Iserlohn, Germany) were used for optical characterization.

This technological sequence resulted in good quality, uniform and homogeneous thin films of ZnO and Mg-doped ZnO.

The crystallization evolution with annealing was studied by X-Ray diffraction (XRD). The crystalline phases of the thin films were identified by a Bruker D8 XRD diffractometer (Bruker AXS GmbH, Karlsruhe, Germany) using a Cu anode (λ_Kα_ = 1.54056 Å), at a grazing angle 2° with a time step of 8 s. Vibrational properties were investigated by FTIR spectroscopy using a Shimadzu FTIR IRPrestige-21 Spectrophotometer (Shimadzu, Kyoto, Japan) in the spectral range 350–4000 cm^−1^ with a resolution of 4 cm^−1^. The bare Si wafer was used as a background. Optical transmittance and reflectance spectra were obtained using a UV–VIS–NIR Shimadzu 3600 double-beam spectrophotometer (Shimadzu, Kyoto, Japan) in the spectral region of 240–1800 nm with a resolution of 0.1 nm and an accuracy of <0.1%. The transmittance was measured against air. The reflectance was taken by using the specular reflectance attachment (5° incidence angle) and the Al-coated mirror as reference. The film thickness of the studied sol–gel films was measured using a LEF 3 M laser ellipsometer equipped with a He–Ne laser operating at a wavelength of 638.2 nm. The LEF 3 M laser ellipsometer is manufactured by Siberian Branch of Russian Academy of Sciences, Novosibirsk. The film morphology of ZnO and ZnO:Mg films was observed by field emission scanning electron (FESEM) microscopy (Philips XL 30FEG-ESEM, FEI). The investigated samples were coated with gold before microscopic observation. The sheet resistance of ZnO:Mg films (on quartz substrates) was measured by four-point probe method using a VEECO instrument, model FPP-100.

## 3. Results and Discussions

### 3.1. XRD Study—Crystalization Behavior

XRD patterns of ZnO and ZnO:Mg films, annealed at temperatures of 300–600 °C, are presented in Figure 1. The studied films are found to be polycrystalline in nature. ZnO films exhibit intense and sharp XRD lines, assigned to the wurtzite structure (Figure 1a). The film crystallization is enhanced with increasing annealing temperatures. The peaks of the sol–gel ZnO:Mg films annealed at 300–600 °C are indexed to the Miller indices (100), (002), (101), (102), (110), (103) and (112), and they match the JCPDS card 01-070-8070 hexagonal wurtzite structure of zinc oxide with space group P63mc (186). ZnO:Mg films reveal an enhancement of the film crystallization with the higher temperature treatments (Figure 1b–e). Mg-doped samples treated at 500 and 600 °C reveal stronger and sharper XRD lines. 

By increasing the annealing temperatures, a new weak diffraction feature is detected at a diffraction angle of 42.8° in the XRD patterns of ZnO:Mg 3 films (500 and 600 °C annealed, Figure 1e, inset) along with the lines attributed to the wurtzite phase. This line at 42.8° corresponds to the (200) plane of cubic MgO (JCPDS card 78-0430, [38]). It must be noted that ZnO:Mg films with lower Mg concentrations reveal no impurity peaks related to Mg and Mg oxide fractions and indicating that the doping of Mg ions does not alter the wurtzite structure. However, the probability of the cluster formations or the existence of Mg-based amorphous phases small enough not to be exposed by XRD measurements cannot be ruled out. The highest Mg concentration (3 wt%) results in a segregation of MgO, although the XRD peaks are very weak (Figure 1e). Similar observations of the coexistence of two phases (wurtzite ZnO and cubic MgO) have been reported by other researchers [39]. The phase separation of MgO and ZnO can be explained by the fact that Mg atoms are more active than Zn atoms and they react with oxygen preferentially [40].

The film crystallization deteriorates with increasing Mg concentrations. It is observed that with increasing magnesium concentrations, the intensity of the XRD peaks decreases, as seen from Figure 1 and Figure 2.

Figure 2a,b exhibits the intensities of the three main XRD lines, 100, 002 and 101, of ZnO and ZnO:Mg films, annealed at the lowest and highest annealing temperatures. The 002 peak intensity is best expressed after thermal treatment at 600 °C and it is reduced with Mg doping (Figure 2b). The most intense XRD line is found to be the (101) peak at a temperature of 400–600 °C of ZnO:Mg films. The diffraction patterns of ZnO:Mg films show the shifting of the 002 peak towards higher diffraction angles (see Figure 2c), which confirms the incorporation of Mg into the ZnO matrix [41,42,43]. This proves that Mg^2+^ ions (with a radius of 0.65 Å) substitute Zn^2+^ ions with a larger radius of 0.74 Å in the wurtzite lattice. It has been observed that the relative intensities of the main XRD peaks ((100), (002), (101)) of the wurtzite phase vary with Mg doping concentration.

The average crystallite sizes are determined by using the peaks (100), (002) and (101) through the Debye–Scherrer’s equation [41,44]:(1)$$D=\frac{k\lambda }{\beta_{2\theta }\cos\left(\theta \right)}$$
where *D* is the crystallite size in nm, *k* is a coupling constant (=0.9), *λ* is the wavelength of the X-ray (*λ* = 1.54056 Å), *β*_2*θ*_ is the FWHM and *θ* is the Bragg angle.

The obtained data of the crystallite sizes are given in Figure 3. The crystallite sizes of pure ZnO films become greater with increasing the annealing temperatures from 9.1 nm (300 °C) to 29.7 nm (600 °C). The crystallites of the doped samples also grow bigger with annealing, revealing a strong dependence on the Mg concentration. The MgO phase, detected in the XRD patterns of ZnO:Mg (Sol 3, greatest Mg amount) films treated at 500 and 600 °C possess crystallite sizes of 10.7 nm (500 °C) and 14.9 nm (600 °C), respectively. It is established that the crystallite sizes are reduced with Mg doping. It is worth noting that ZnO:Mg 2 films have the smallest crystallites.

The decrease in the crystallite sizes with Mg doping might be caused by a Zener pinning effect, i.e., the prevention of motion of grain boundaries either by precipitation of secondary phases or by contamination at the surface [45]. An additional MgO phase may act as an obstacle for the growth of crystallites in ZnO:Mg 3 films. It is also observed that the crystallites grow with different sizes along the (100), (002), (101) diffraction planes. Figure 4 presents the estimated sizes of the crystallites, estimated from XRD data for the dependence of the three strongest lines on Mg concentration and annealing temperatures. It can be seen that the crystallites growing along the (002) plane become greater than the crystallites growing along other two diffraction planes at 500 °C (Figure 4c) for all studied films. After the highest annealing temperature (600 °C), crystallites grown along the (002) plane are greater for ZnO, ZnO:Mg 0.5 and ZnO:Mg 1. ZnO:Mg 3 films show slightly smaller sizes for the crystallites grown along the (002) plane.

The texture coefficient of a plane illustrates the texture of a particular plane and deviation from the standard sample implies the preferred growth. The texture coefficient (*TC* (*hkl*)) can be calculated from XRD patterns using [46]:(2)TC hkl=IhklIohklN−1∑nIhklIohkl
where *I*(*hkl*) and *I_o_*(*hkl*) are the measured relative intensity and JCPDS standard intensity, respectively, of a plane (*hkl*); *N* is the reflection number; and *n* is the number of diffraction peaks. 

If there is a preferential orientation, the TC (hkl) value should be greater than one. If *TC* (*hkl*) is equal to or less than 1, then the sample is proposed to have randomly oriented crystallites. The texture coefficient was determined for the sol–gel ZnO and Mg-doped ZnO films using Equation (2). TC values were calculated for the diffraction planes (100), (002) and (101) considering the measured and standard (JCPDS card 01-070-8070) intensities of all detected peaks. The degree of the preferred orientation for three diffraction planes is given in Figure 5. ZnO-based films demonstrate a preferential growth along the (002) plane as the TC values of ZnO and ZnO:Mg films are greater than 1 (see Figure 5a). For example, TC (002) values of ZnO films are increased up to 2.46 with increasing annealing temperatures, along with a decrease of the TC (100) and TC (101) values. On the other hand, ZnO:Mg films also reveal the highest values for TC (002) in comparison to those for TC (100) and TC (101), but the texture coefficients strongly depend on Mg concentration and thermal treatments. TC (002) values of ZnO:Mg 0.5 and ZnO:Mg 1 films are increased with the annealing temperatures, in contrast to other doped films with higher Mg contents, where the TC (002) values are slightly smaller after high temperature treatments. TC (002) shows a slight increase with Mg doping for 300 °C annealed samples (1.6–1.8) (except the ZnO:Mg 3 films). After thermal treatments at 400 and 500 °C, ZnO:Mg films manifest a well pronounced decrease in the texture coefficient in the (002) plane, and 600 °C treatment has the greatest impact on TC (002) and it drops sharply with increasing Mg doping. All studied films manifest a decrease in the texture coefficient of the (101) plane with a change in the annealing temperature from 300 to 600 °C (Figure 5b). The lowest TC (101) values are determined for undoped ZnO films and the highest values are estimated for ZnO:Mg 3 films. TC (100) is reduced with annealing for all studied films as the lowest values are found for undoped ZnO, as can be noticed from Figure 5c. ZnO:Mg 3 films exhibit higher values for TC (100) compared with the other studied films.

To gain further insight, the lattice parameters *c* and *a*, the *c/a* ratio, Zn–O bond length and *u* parameter of the sol–gel ZnO and ZnO:Mg films wre determined by Equations (3)–(5) [46] using the crystallographic plane indices (*h*, *k*, and *l*). The lattice parameters were estimated using the X-ray diffraction least squares unit cell refinement spreadsheet templates [47]. The obtained results are given in Figure 6 and Figure 7.
(3)$$\frac{1}{d_{hkl}^{2}}=\frac{4}{3}\left(\frac{h^{2}+hk+k^{2}}{a^{2}}\right)+\frac{l^{2}}{c^{2}},\;a=b=\frac{\lambda }{\sqrt{3}\sin\theta },\;c=\frac{\lambda }{\sin\theta }$$
(4)$$L=\sqrt{\left(\frac{a^{2}}{3}\right)+{\left(\frac{1}{2}-u^{2}\right)}^{2}c^{2}},$$
where *u* is the internal parameter:(5)$$u=\left(\frac{1}{3}\right)\left(\frac{a^{2}}{c^{2}}\right)+\frac{1}{4}$$

The lattice parameters, *a* and *c*, are changed, which was induced by Mg doping in the ZnO structure. It is revealed that the *a* axis is elongated and the *c* axis is shortened (Figure 6a). This effect is due to the fact that zinc ion (0.60 Å) sites can be occupied by smaller Mg ions (0.57 Å) in the ZnO host matrix. Similar results for the behavior of the lattice parameters for ZnO:Mg films and nanoparticles can be found in the literature [42,48]. It must be noted that ZnO:Mg 2 films manifest the greatest change, followed by ZnO:Mg 3 (highest Mg concentration). Wurtzite ZnO (JCPDS card 01-070-8070) has standard values of lattice constants *c* = 5.2049 Å and *a* = 3.2489 Å.

The characteristic of the hexagonal lattice is the *c/a* ratio. ZnO has hexagonal wurtzite structure with a hexagonal unit cell and two lattice parameters *a* and *c*. The *c/a* ratio is known to be 1.633 for the ideal arrangement [49]. Generally, the real lattice of ZnO thin films differs from the ideal structure due to deposition methods, the presence of impurities (concentration of doping elements, defects and the difference of ionic radii with respect to the substituted matrix ion), and external factors, such as external strains, temperature and others [50]. The *c/a* ratio (Figure 6b) is affected by high thermal treatments and Mg incorporation. XRD analysis shows that undoped ZnO films are already substantially distorted, as indicated by the deviation of the calculated *c/a* ratio (1.6063 to 1.6017) from that of the standard geometry 1.633. It is observed that the magnesium substitution results in a significant deformation of the wurtzite structure, with a more pronounced effect for the higher doping concentrations and the higher annealing temperatures. The deviations from the ideal value are due to the defects in the crystal lattice structure. The *c/a* ratio significantly decreases with Mg concentration and the greatest deviation is found for ZnO:Mg 2 films. The differences between *c/a* ratio values of sol–gel ZnO and ZnO:Mg films and the ideal hexagonal ZnO arrangement result from the existing defects in the crystal lattice and the quantity of defects are dependent on crystallite sizes, the dopant effect and dopant concentration. The results reveal that magnesium doping in the zinc oxide matrix leads to a less close packing in the hexagonal crystalline structure.

The variation of the bond length Zn–O and parameter *u*, defined as the positional parameter of the wurtzite structure, with Mg incorporation and annealing temperatures is given in Figure 7. It is known that the Zn^2+^ ionic radius is 0.83 Å and the ionic radius of O^2−^ is 1.38 Å. The bond length of Zn–O is expected to be 2.21 Å. The smaller values determined for ZnO and ZnO:Mg films (Figure 7a) are an indication of the presence of structural defects (especially oxygen vacancies). The *u* parameter is 0.375 in an ideal wurtzite structure. In our case, the *u* parameter is varied from 0.379 to 0.381, depending on Mg concentration and annealing. These values are close to the reported data for the hexagonal wurtzite Mg_x_Zn_1−x_O [51].

The doping of metal ions into ZnO can generate both a lattice distortion and a tendency to create lattice defects and nucleation centers [29,30]. The crystal lattice distortion degree, *R*, is determined according to the equation in ref. [52]:(6)R=2a2312c

The distortion degree represents the deviation of the crystal from a perfect arrangement, and it is accepted that *R* = 1 suggests that there is no distortion [52]. The dependence of *R* on the annealing temperatures is shown in Figure 8 for ZnO and ZnO:Mg films. The degree of distortion of the sol–gel films is greater than 1 and magnesium doping causes increasing the distortion degree, *R*, with the greatest distortion of the crystal lattice being established for ZnO:Mg 2 films. On the other hand, a non-linear increment of *R* values is observed with annealing temperatures ranging from 300 to 600 °C. It can be suggested that the thermal treatments introduce new defects into the ZnO-based films and, respectively, induce higher crystal imperfection.

From XRD analysis, it can be concluded that ZnO and ZnO:Mg films have a polycrystalline nature with wurtzite being the predominant phase. The film crystallinity is deteriorated with increasing Mg doping concentration. Since the diffraction peaks become weaker and wider with the increase in Mg^2+^ concentration, magnesium doping inhibits the growth of ZnO nanocrystals and affects the crystallization of the films. The crystallite sizes of ZnO:Mg films are reduced. The evolution of the lattice parameters, *c/a* ratio, Zn–O bond, *u* parameter and distortion *R* was studied as a function of Mg concentration and thermal treatments. The MgO phase co-exists with wurtzite ZnO in the structure of ZnO:Mg 3 films, annealed at 500 and 600 °C. For the other doped films, no Mg or Mg oxide containing phases are observed in the frames of XRD detection limits. It is worth noting that ZnO:Mg 2 films show some differences compared with other doped ZnO films, as these films manifest the smallest crystallites values, the largest shift of the 002 peak position, and the biggest variations in the lattice parameters and distortion degree. 

### 3.2. FTIR Spectroscopy—Vibrational Properties

To further study the impact of the Mg dopant, FTIR spectroscopy was applied as a sensitive characterization method supplementary to the XRD investigation. FTIR spectroscopy is used for revealing the vibrational modes and functional groups in the studied ZnO and ZnO:Mg films. It is well known that the positions and intensities of the absorption peaks depend on the chemical composition, structure (crystallinity, particle sizes and shapes) and morphology of thin films [53,54]. Figure 9 presents FTIR spectra of ZnO and ZnO:Mg films, annealed at temperatures of 300–600 °C. The water and hydroxyl group stretching vibrations are mainly revealed in the spectral range 3000–4000 cm^−1^, where FTIR spectra of ZnO and Mg-doped ZnO films manifest absorption features. ZnO films manifest a very broad and strong absorption band at 3420 cm^−1^, extended over 3190–3600 cm^−1^ after 300 °C treatment, and this band disappears with increasing annealing temperatures. ZnO:Mg films also reveal broad absorption bands at 3395 cm^−1^ (3130–3600 cm^−1^), which depend on the Mg content and the thermal treatments.

The strongest absorptions in this spectral range are observed for ZnO:Mg 3 films and, although their intensity decreases at the higher temperatures, they are still noticeable in the spectra. For the other Mg-doped ZnO films, the band disappears after 400 °C treatment. Another weak IR line can be detected at 3740 cm^−1^ for all studied films. ZnO:Mg 0.5, 1 and 2 films also show weak absorption lines at 3850 cm^−1^, independent of the thermal treatments. The spectral range 3000–4000 cm^−1^ is indicative of the stretching vibrations of hydrated species. The band at 3850 cm^−1^ is assigned to hydroxyl groups, bound to two deficient surface sites. The absorption band at 3740 cm^−1^ is related to the vibrations of the separate OH groups [55]. The broad bands at 3420 cm^−1^ (3190–3600 cm^−1^ for 300 °C ZnO) and at 3395 cm^−1^ (3130–3600 cm^−1^ for ZnO:Mg) are clearly overlapping, namely due to the hydrogen bound OH group (around 3400 cm^−1^), OH groups, related to Zn vacancies (around 3220 cm^−1^) and/or physically adsorbed water [56,57,58]. The clear absorption band at 1600 cm^−1^ is attributed to the bending OH vibrations of adsorbed water [56]. The weak lines at 2360 and 2325 cm^−1^ (seen in all measured spectra) are due to atmospheric CO_2_ molecules, as FTIR measurements are performed in air [59]. The weak lines in the range 1300–1400 cm^−1^ are attributed to C–O stretching modes [60]. These lines are not observed for the films treated at higher temperature.

For ZnO:Mg 2 and 3, there is a clear broad absorption band around 1440 cm^−1^, which becomes better defined after 500 and 600 °C annealing (see Figure 9c,d). IR lines in the FTIR spectra of undoped ZnO films around this spectral range are completely missing. Some authors claim that these absorption bands are related to Mg–O stretching vibrations [61,62]. This is an indication of Mg incorporation into ZnO. It must be noted that XRD analysis revealed that the ZnO films with the highest Mg concentration (ZnO:Mg 3) after high temperature treatment at 500 and 600 °C possess a small fraction of the cubic MgO phase. FTIR analysis suggests that the MgO phase might have also formed in the structure of ZnO:Mg 2 films. The lines at 1100 cm^−1^ (only for the spectra of ZnO:Mg 2) and at 1020 cm^−1^ (ZnO and ZnO:Mg 2, treated at 300 °C and ZnO:Mg 3, annealed at all temperatures) are attributed to the stretching modes of C–O [31], due to carbon impurities.

The spectral range below 1000 cm^−1^ is very important for studying characteristic metal–oxygen frequencies, related to Zn–O and Mg–O bonds and other IR lines attributed to the functional groups. Undoped ZnO films reveal a very weak line at 667 cm^−1^, a broad absorption band at 617 m^−1^, and clear IR lines at 520, 490, 472 and 360 cm^−1^. All these lines are assigned to Zn–O stretching modes [63] as the absorption feature at 520 cm^−1^ is related to Zn–O bonding in the wurtzite crystal structure [64]. The main absorption band of sol–gel ZnO films is intense and broad, which becomes stronger with increasing the annealing temperature. It shows a shift from 410 to 405 cm^−1^ with thermal treatments. This IR band is also related to Zn–O stretching vibrations in the wurtzite ZnO structure [64].

Figure 9a presents FTIR spectra of ZnO and ZnO:Mg films, annealed at 300 °C. It can be observed that the main absorption bands of ZnO:Mg films are broader than the corresponding ZnO features. Interesting behavior is recorded for ZnO:Mg 2. The FTIR spectrum manifests a splitting of the main absorption feature into three strong lines, positioned at 374, 394 and 419 cm^−1^. Such splitting is found in all FTIR spectra of ZnO:Mg 2 films. The spectrum of ZnO:Mg 3 films reveals clear bands at 472 and 515 cm^−1^, and weaker bands at these locations can be seen in the spectra of ZnO:Mg 1 and ZnO:Mg 2 films. With increasing annealing temperature (400 and 500 °C), the main absorption bands of ZnO:Mg films differ from that of undoped ZnO (Figure 9b,c). The absorption main band of ZnO:Mg 2 is split. The spectra of 600 °C treated samples (Figure 9d) show that the main bands are broader for Mg-doped ZnO films (as can be seen from the inset image). The clear peak at 470 cm^−1^ is observed for ZnO and ZnO:Mg films. The spectra exhibit IR lines at 470 and 515 cm^−1^ (for ZnO:Mg 0.5 and ZnO:Mg 1) and at 460 and 515 cm^−1^ (ZnO:Mg 2 and ZnO:Mg 3). As it has been mentioned above, the IR lines at 420 and 522 cm^−1^ are attributed to the stretching vibrations of Zn–O in the ZnO wurtzite structure [64].

The Mg–O vibrations are reported to be located at 449, 511, 584 and 671 cm^−1^ [65] and at 771, 476 and 430 cm^−1^ [66]. Other Mg–O–Mg vibration modes of MgO can be found at 880 cm^−1^ [67]. At the lowest temperature treatment, some of the peaks between 400 and 520 cm^−1^, appearing in the spectra of Mg-doped ZnO films, can correspond to contributions of Mg–O vibrational modes [68], but it must be considered that different Zn–O stretching vibrations are also detected in this spectral range. FTIR analysis reveals changes in the positions, shapes and intensities of the absorption bands, which are a distinct indication of Mg incorporation into the ZnO host lattice. Some IR lines can be attributed to Mg–O bonds, which can especially be confirmed for ZnO:Mg 2 and ZnO:Mg 3 films. These conclusions support XRD interpretations that the MgO phase exists in ZnO:Mg 3 films. FTIR analysis proposes that the Mg oxide fraction is presented in ZnO:Mg 2 films. As previously discussed, these films exhibit the greatest differences in crystallite sizes, lattice parameters and distortion degrees, proved by XRD analysis.

### 3.3. UV–VIS Spectroscopy—Optical Characterization

Figure 10 shows the transmittance and reflectance spectra of the ZnO and ZnO:Mg films in the spectral range 240–1000 nm for different annealing temperatures. The substrates used are quartz. The transparency of the sol–gel films strongly depends on the magnesium doping content and the annealing temperature. The specific bands observed in the optical spectra of undoped ZnO films below the bandgap wavelength are due to the excitonic absorption.

The excitonic feature of the bulk ZnO is located near 373 nm. The excitonic absorption of the studied ZnO films is expressed at 345 nm. ZnO:Mg 0.5 and ZnO:Mg 1 films also manifest excitonic absorption at 335 nm (Figure 10c,d). The films with higher Mg content show no such absorption features in the transmittance spectra. The excitonic band is related to the film crystallinity and its presence reveals that the samples are highly crystalline [69]. 

Figure 11 presents the average transmittance (T_average_) and reflectance (R_average_) values in the visible spectral range 450–750 nm for the sol–gel films, treated at different annealing temperatures. The transmittance is measured against air. The average transmittance of the bare quartz substrate is 93.5% and the average reflectance is 6.7%. The most transparent films are found to be ZnO:Mg 2 samples and ZnO:Mg 3 films exhibit the lowest transmittance values, respectively. The low transmittance of ZnO:Mg 3 films can be related to MgO phase segregation, as has been observed by other authors [70]. ZnO:Mg 1 and ZnO:Mg 2 films possess improved optical transparency compared to other sol–gel ZnO films. Mg incorporation leads to improved film transparency up to 500 °C annealing with the exception of the highest Mg doping (3 wt% Mg, ZnO:Mg 3 films). The transmittance of the doped films can increase due to smaller crystallite sizes, decreasing the light scattering, respectively. The crystallites sizes of ZnO:Mg 2 films are the smallest and their film transparency is the highest, respectively.

The average reflectance of undoped ZnO films is below 5% (Figure 11b). The reflectance of ZnO:Mg films is higher and is strongly influenced by the doping concentration and annealing. ZnO:Mg 0.5 films have R_average_ < 7%. The reflectance of ZnO:Mg 1 and ZnO:Mg 2 films increases with annealing temperatures to 23.6 and 15.8%, respectively. ZnO:Mg 3 films show an increase from 7.5 to 22.2% up to 500 °C annealing, and then a drop to 16.3% after 600 °C treatment.

Tauc relation is employed to estimate the direct optical band gap of the sol–gel films [70,71]:(*αhν*) = *B* (*hν* − *E_g_*)^1/2^(7)
where ‘*B*’ is a constant, *h* is Plank’s constant, *ν* is the photon frequency, α is the absorption coefficient and *E_g_* is the optical band gap. The absorption coefficient *α* was estimated using the film thickness *d* and the transmittance T [8]:(8)$$\alpha =-\frac{1}{d}In\left(\frac{1}{T}\right)$$
where *d* is the film thickness. The film thickness of undoped ZnO is 190 nm. Mg doping results in a slight increase in the film thickness. ZnO:Mg 0.5 films are 194 nm thick. The thickness of ZnO:Mg 1 and ZnO:Mg 3 films are 195 and 197 nm, respectively. ZnO:Mg 2 films possesses the greatest film thickness of 210 nm. The optical band gap (*E_g_*) values were determined from Equations (7) and (8) and are given in Figure 12. The obtained results clearly reveal that Mg incorporation enlarges the optical bandgap. ZnO films have *E_g_* values ranging from 3.27 to 3.24 eV with increasing temperatures from 300 to 600 °C. 

The shrinkage of the optical band gap with annealing temperature can be due to the improved crystallization, the decreased defects of the film structure and greater crystallite sizes. Similar results have been reported in the literature [71].

Mg-doped films exhibits a widening of the optical band gap with respect to undoped ZnO films. ZnO:Mg 2 films possess the highest values of E_g_ and the optical band gap reduces sharply after thermal treatment at 600 °C. A similar tendency is found for ZnO:Mg 3. The optical band gap varies from 3.32 eV (for ZnO:Mg 0.5, 300 °C) to 3.56 eV (ZnO:Mg 2, 500 °C). The E_g_ values of sol–gel ZnO and ZnO:Mg films are in the range of the reported data for these materials [72]. The enhanced band gap can be induced by several factors: (i) the reduced lattice constant *c* [73], whreby the XRD study confirms that the lowest *c* parameter is found for the sol–gel ZnO:Mg 2 films; (ii) the difference in electronegativities of Zn and Mg atoms may induce an attractive effect of interaction between Mg^2+^ and O^2−^, which may lead to a rise in band gap energy as MgO possesses a band gap energy of 7.8 eV [74] and the possible existence of the MgO phase in the structure of doped films can induce a wider band gap [75]; (iii) the Moss–Burstein effect is also possible, where Mg ions disturb the ZnO lattice leading to oxygen vacancies and to a shift of the Fermi level towards the conduction band—blue shifting [76]; and (iv) the decrease in the crystallite sizes [77].

The refractive index *n* is an important parameter for materials used as optical elements and applied in integrated optic devices and optoelectronic devices. It is well known that refractive index is one of the fundamental properties for an optical material and it is closely related to the electronic polarizability of ions and the local field inside materials [78].

The extinction coefficient and the refractive index of ZnO and ZnO:Mg films can be determined from the spectrophotometric data, using the equations [79]:(9)$$k=\frac{\lambda \alpha }{4\pi },$$
(10)$$n=\frac{\left(1+R\right)}{\left(1-R\right)}+\sqrt{\frac{4R}{{\left(1-R\right)}^{2}}-k^{2}},$$
where *R* represents the film reflectance, *k* is the calculated extinction coefficient, *λ* is the wavelength of the incident photon and *α* is the absorption coefficient, estimated from Equation (8).

The comparison of the extinction coefficient of ZnO and Mg-doped ZnO films is given in Figure 13 for annealing temperatures from 300 to 600 °C. The spectral range is 300–1000 nm. The extinction coefficient increases with the annealing temperatures and ZnO:Mg 3 films show the highest values in the visible spectral range 450–750 nm after annealing at 300–500 °C. ZnO:Mg 0.5 and ZnO:Mg 2 have the lowest extinction values in the visible spectral region. ZnO and Mg-doped ZnO films reveal that the extinction coefficient is increased with raising the annealing temperature.

The estimated values of the refractive index are presented in Figure 14, where the obtained results of ZnO and ZnO:Mg films are compared at different annealing temperatures. The refractive index of the undoped ZnO films vary from 1.25 to 1.90 in the range 450–750 nm, depending on the wavelength *λ* and the annealing temperature. It is revealed that the *n* values become greater increasing annealing temperature. The refractive index is increased with increasing wavelength. Similar behavior of the refractive index for ZnO thin films has been reported by others [80].

Mg doping leads to a higher refractive index compared to undoped ZnO films, as can be noticed from Figure 14. ZnO:Mg films also show a strong dependence of the refractive index on the annealing temperatures, except for ZnO:Mg 1 samples. The higher refractive index of the doped ZnO films can be related to the formation of agglomerations resulting from Mg doping. It is also well known that the refractive index is sensitive to structural defects and depends on dopants, impurities, crystallinity, deposition methods, stoichiometry, packing density, etc. [78,81].

A similar increase in the refractive index with doping with different elements is reported for sol–gel boron-doped ZnO films [82] and for Fe, Co, Mn, Ni doped ZnO films deposited by spray pyrolysis [81]. The refractive index values are also dependent on the experimental optical measurement technique used (spectrophotometry or spectroscopic ellipsometry) and the approaches used for extraction of the refractive index from these measurements [78]. Table 1 presents a small part of the reference data for *n* values of ZnO materials, prepared by different methods [6,78,79,81,82,83,84,85,86,87,88,89,90,91]. It can be seen that the refractive index values vary in a very broad range. Mg doping also results in an increase in the refractive index of ZnO, as it has been determined that the doping concentration and the annealing temperatures significantly affect the *n* values.

The optical study of the sol–gel ZnO films reveals that their optical properties are affected by Mg doping. For a certain dopant content, the transparency of the films is improved. The optical band gap can be tuned by varying the Mg concentration in the sol solution. The refractive index is changed with doping and annealing.

### 3.4. FESEM Microscopy—Film Morphology

The study of the surface morphology was performed using FESEM for undoped ZnO, ZnO:Mg 0.5 and ZnO:Mg 3 films, annealed at 600 °C (Figure 15 and Figure 16). The substrate used was Si wafer. FESEM images with different magnifications are used for the film morphology analysis.

In our previous research on ZnO-based films [37], we used the same sol solution for depositing pure ZnO films, at a higher spin coating rotation speed (4000 rpm). In this work, ZnO and ZnO:Mg films were deposited at a lower spin coating speed of 2000 rpm. It is interesting to inspect the impact of the rotation speed change on the film morphology. FESEM micrographs of the sol–gel ZnO films, deposited at 2000 and 4000 rpm and annealed at 600 °C, are given in Figure 15.

The images reveal a wrinkle network structure with irregular fibers (magnification 20,000 and spatial resolution of 1 micron), and in closer observation (80,000, 300 nm, imbedded figures) they reveal porous surfaces. Some differences in the film morphologies appeared: a smoother surface is observed for the ZnO film, obtained at 2000 rpm compared to other films. Its surface shows the presence of spherical particles, as well as irregular grains with different shape and sizes. The ZnO film, deposited at 4000 rpm exhibits a more porous structure and a rougher wrinkle network. Similar observations of wrinkle network morphology have been previously reported for sol–gel spin-coated ZnO films [92].

The wrinkle formation of sol–gel films can emerge due to the following reasons: (1) the increase in volumetric stress in the films and evaporation of the solvent [93], (2) the lack of hydroxyl (or alkoxy) groups in the sols [94] or (3) the release of mechanical stresses developed during densification and heat treatment of the films [92]. Other probable factors can be differences between the thermal expansion coefficients of the substrate and film, volume reduction upon crystallization and film thickness [92]. The undoped ZnO films (independent on the spin coating speed) exhibit weak and broad absorption bands due to the stretching vibrations of hydroxyl groups (at 3190–3600 cm^−1^) after 300 °C annealing, which disappeared at higher annealing temperatures. 

Figure 16 presents FESEM images of sol–gel ZnO, ZnO:Mg 0.5 and ZnO:Mg 3 films, annealed at 600 °C. The magnifications used are 10,000 with a spatial resolution of 2 microns and 160,000 with a spatial resolution of 200 nm. It can be confirmed that Mg doping changes ZnO film morphology as the wrinkle formation is distorted.

Undoped ZnO film exhibits a wrinkle-type network (as discussed above) and a relatively smooth surface. The higher magnifications reveal a close-packed grained structure with spherical and irregular particles. The wrinkles have widths of 190–540 nm and the grain sizes are in the range of 30–120 nm, but the greater particles prevail. The lowest Mg concentration (0.5 wt%) changes the film morphology (Figure 16c,d): the wrinkle-like network still exists, but with a low density of wrinkles and fiber-like streaks, indicating a smoother surface. The closer view manifests the presence of voids (porous structure). Spherical particles and irregular grains are observed, where the smallest particles are 25–30 nm, and the greater particles are up to 80–90 nm, with a small quantity of greater grains up to 120–170 nm. Increasing Mg concentration (ZnO:Mg 3 films) results in a smooth but rather porous surface (Figure 16e,f) with no wrinkles and fibers. Again, the spherical particles are predominantly grown. The smallest grains are 30–44 nm with some bigger particles being up to 70–90 nm; grains with sizes close to 100 nm and slightly greater are exceptions and are rarely seen. ZnO films show bigger grains as has been deduced by XRD analysis.

The film morphology of ZnO films is affected by Mg doping. The wrinkles are smaller and almost disappear for the highest magnesium concentration. The smoother film surface leads to an improved transparency of ZnO:Mg films, especially for ZnO:Mg films with 0.5, 1 and 2 wt% Mg concentration.

### 3.5. Figure of Merit (FOM)—Electrical Properties

Sheet resistance (R_sheet_) of sol–gel ZnO and ZnO:Mg films is measured by the four-point probe technique. Optical transparency and sheet resistance are two factors that are crucial for transparent conductive electrodes based on metal oxides and are essential for applications of these materials in solar cells [95]. Optical transparency directly governs the amount of charge generation within solar cells. The influence of sheet resistance is more complicated and less straightforward, but it also strongly impacts the device’s performance [95].

A figure of merit is defined as an appropriate quantitative measure that can rate the performance of the sample and determine its comparative effectiveness for an application. Figure of merit (FOM) can be estimated from the following equation [96]:(11)$$FOM=\frac{T_{average}^{10}}{R_{sheet}}$$
where *T_average_* is the average transmittance in the spectral range 450–750 nm and *R_sheet_* is the measured sheet resistance. The obtained values are given in Table 2. 

Table 3 presents a comparison of the FOM and resistance values found in the literature for ZnO-based thin films [96,97,98,99,100].

It can be observed that the FOM of the studied films are lower or similar to reported values of ZnO-based materials. The electrical properties of the undoped and Mg-doped ZnO films are affected by Mg concentration and high temperature treatments. It can be observed that ZnO:Mg 2 films possess the best optical and electrical properties. Further study is needed for optimization of the film thickness, transparency and sheet resistance of sol–gel Mg-doped ZnO films, but the obtained results are encouraging.

## 4. Conclusions

Thin films of ZnO and ZnO:Mg were successfully deposited by the sol–gel spin coating method on silicon wafers and quartz substrates. Their structural, optical and morphological properties were studied as functions of four magnesium doping concentrations and different thermal treatments (300–600 °C). The XRD study proves that Mg doping induces the deterioration of the film crystallizations. MgO segregation is found in ZnO:Mg 3 films with the highest Mg concentration after annealing at 500 and 600 °C. No Mg or Mg oxide phases are detected in the other ZnO:Mg films. The modifications in lattice parameters, *c/a* ratio, Zn–O bond and the shifting of the 002 XRD line corroborates the successful incorporation of magnesium into the ZnO host matrix. The wide absorption bands at 1400 cm^−1^, observed in the FTIR spectra of ZnO:Mg 2 and ZnO:Mg 3 films, are attributed to Mg–O stretching vibrations and are completely absent in the other studied pure ZnO and doped ZnO samples. The FTIR study reveals that a small MgO fraction can also exist in ZnO:Mg 2 films, most probably in the amorphous state. This finding explains the crystalline behavior of ZnO:Mg 2 films, with the smallest values of the crystallite sizes, *c* parameter and *c/a* ratio and, respectively, the greatest distortion parameter. 

Optical characterization reveals that the film transparency of the Mg-doped ZnO films is generally enhanced. The greatest transmittance is found for ZnO:Mg 2 films. Mg doping tunes the optical band gap (E_g_) values and Mg incorporation results in a widening of E_g_. The influence of the annealing temperatures on the transmittance, reflectance, band gap, extinction coefficient and refractive index was monitored. FESEM analysis exhibits that the surface features are strongly dependent on Mg doping. The film morphology is changed from a wrinkle-type surface to a smoother morphology with increasing the magnesium doping.

## Figures and Tables

**Figure 1 materials-15-08883-f001:**
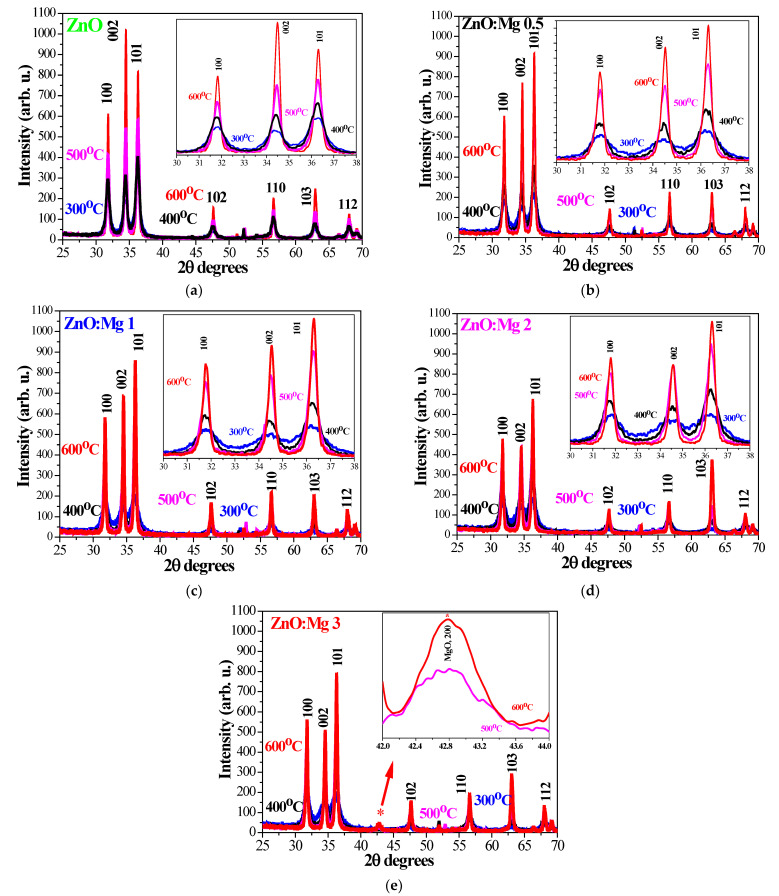
XRD patterns of the sol–gel (**a**) ZnO, (**b**) ZnO:Mg 0.5, (**c**) ZnO:Mg 1, (**d**) ZnO:Mg 2 and (**e**) ZnO:Mg 3 thin films, treated at temperatures of 300–600 °C. The asterisk symbol * indicates MgO. _2_. The inset pictures in (**a**–**c**) are the enlarged patterns in the range of 30–38°. The inset in (**e**) presents the appearance of the MgO phase, 42–44°.

**Figure 2 materials-15-08883-f002:**
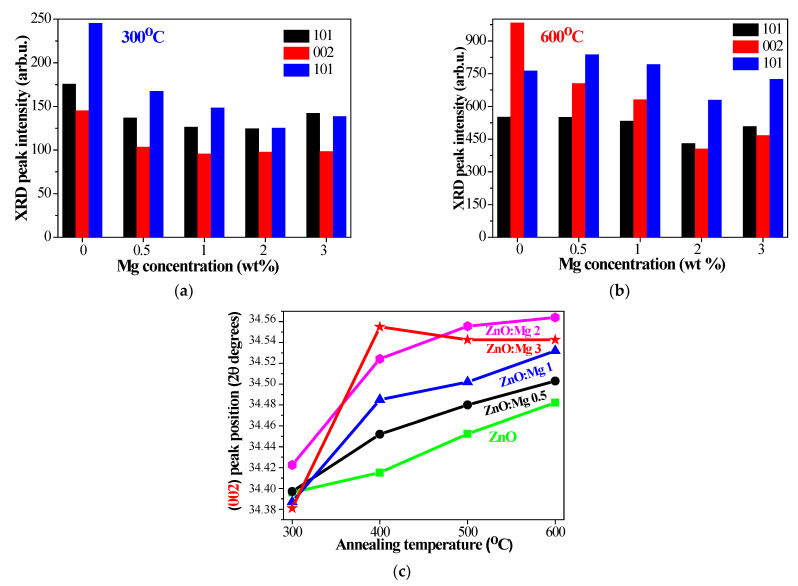
Change of XRD intensities of (100), (002), (101) peaks with Mg concentration, annealed at (**a**) 300 and (**b**) 600 °C. (**c**) Comparison of the diffraction angle shift of the (002) peak of ZnO and ZnO:Mg films, treated at 600 °C.

**Figure 3 materials-15-08883-f003:**
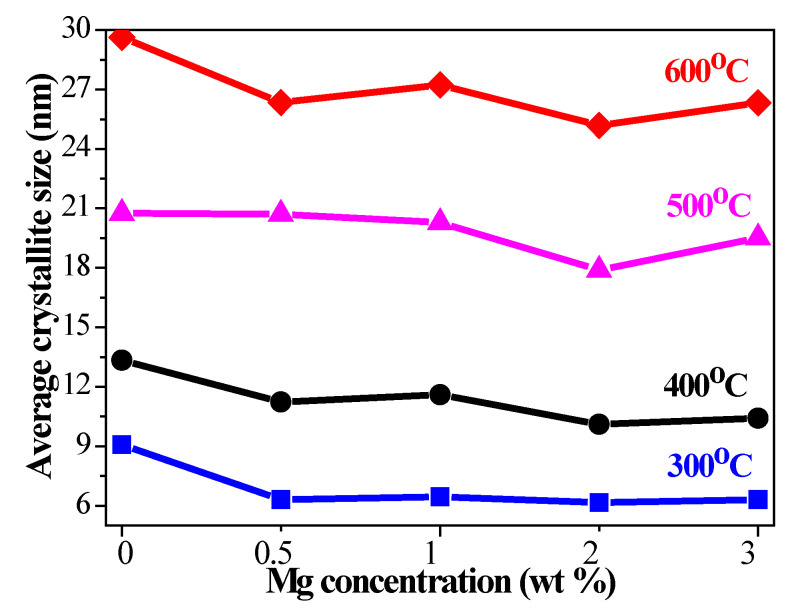
Average crystallite sizes of the sol–gel ZnO and ZnO:Mg films as a function of magnesium concentration. The films are annealed at temperatures from 300 to 600 °C.

**Figure 4 materials-15-08883-f004:**
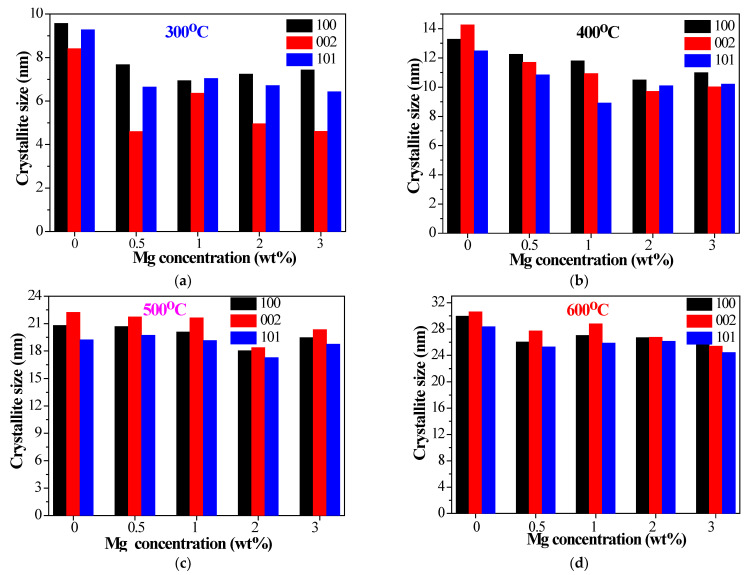
Crystallite sizes along (100), (002), (101) diffraction planes of ZnO and ZnO:Mg films, annealed at (**a**) 300, (**b**) 400, (**c**) 500 and (**d**) 600 °C.

**Figure 5 materials-15-08883-f005:**
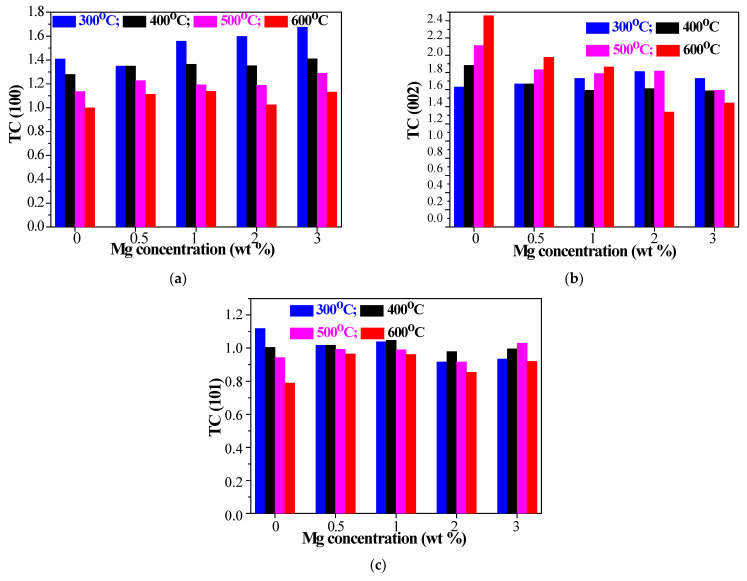
Texture coefficients of ZnO and ZnO:Mg films, depending on the thermal treatments and magnesium doping: (**a**) TC (100), (**b**) TC (002) and (**c**) TC (101).

**Figure 6 materials-15-08883-f006:**
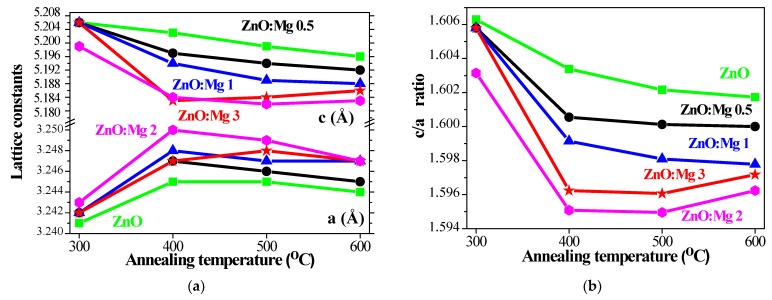
Estimated values of (**a**) the lattice parameters *c and a* and (**b**) the *c/a* ratio as a function of the annealing temperature of ZnO and ZnO:Mg films.

**Figure 7 materials-15-08883-f007:**
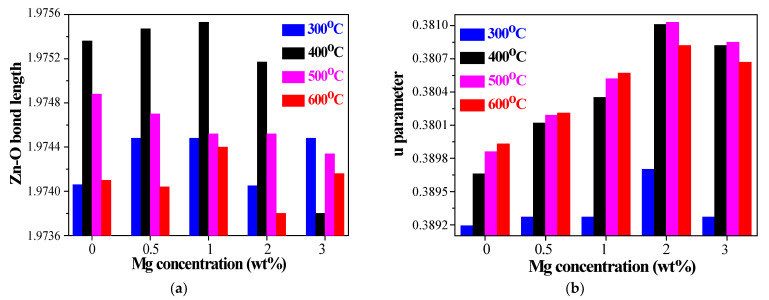
Determined values of (**a**) Zn–O bond length and (**b**) *u* parameter, depending on magnesium concentration of 300–600 °C annealed ZnO and ZnO:Mg films.

**Figure 8 materials-15-08883-f008:**
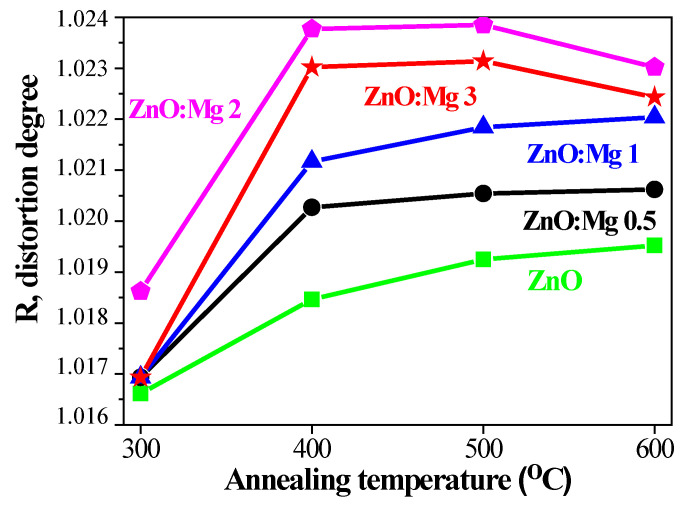
Crystal lattice distortion degree, *R*, estimated for ZnO and ZnO:Mg films depending on annealing temperatures.

**Figure 9 materials-15-08883-f009:**
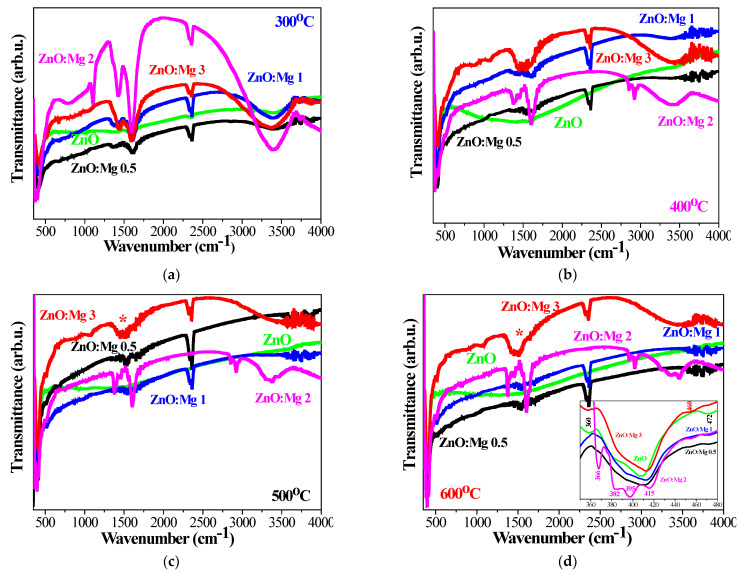
FTIR spectra of sol–gel ZnO and ZnO:Mg films, treated at (**a**) 300, (**b**) 400, (**c**) 500 and (**d**) 600 °C. The inset figure in (**d**) presents FTIR spectra of 600 °C annealed samples in the extended spectral range 350–480 cm^−1^. The asterisk * in (**c**,**d**) denotes the absorption band due to Mg–O vibrations.

**Figure 10 materials-15-08883-f010:**
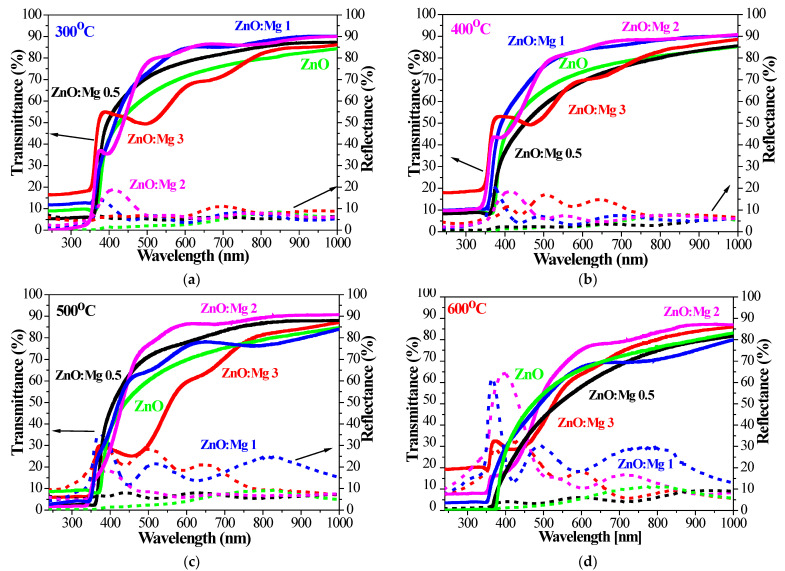
Comparison of the transmittance and reflectance spectra of sol–gel ZnO and ZnO:Mg films, treated at (**a**) 300, (**b**) 400, (**c**) 500 and (**d**) 600 °C. Transmittance of the bare quartz substrate is given as a reference.

**Figure 11 materials-15-08883-f011:**
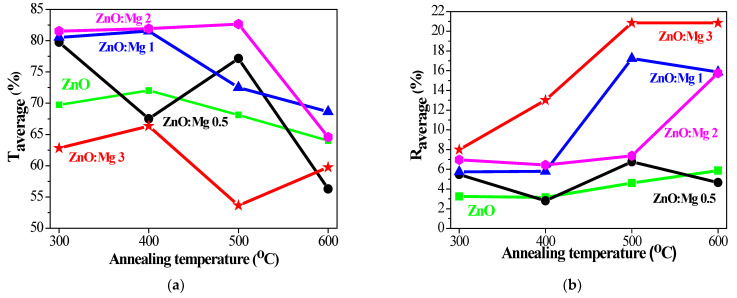
(**a**) Average transmittance (T_average_) and (**b**) average reflectance (R_average_) values estimated for the spectral range 450–750 nm of the sol–gel ZnO and ZnO:Mg films, depending on the annealing temperatures from 300 to 600 °C.

**Figure 12 materials-15-08883-f012:**
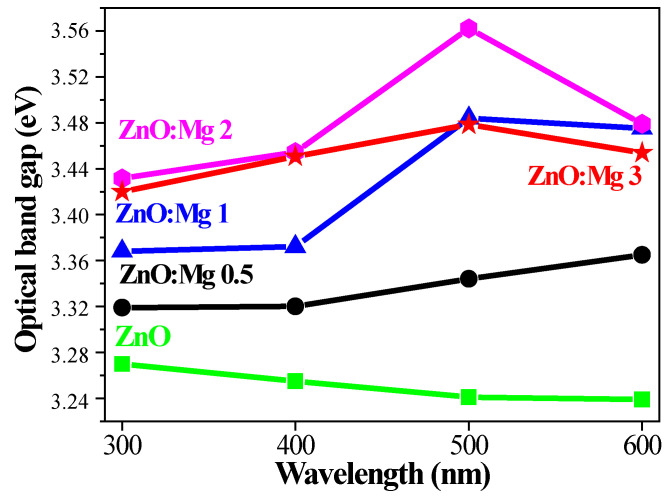
Optical band gap of ZnO and ZnO:Mg films as a function of the annealing temperatures.

**Figure 13 materials-15-08883-f013:**
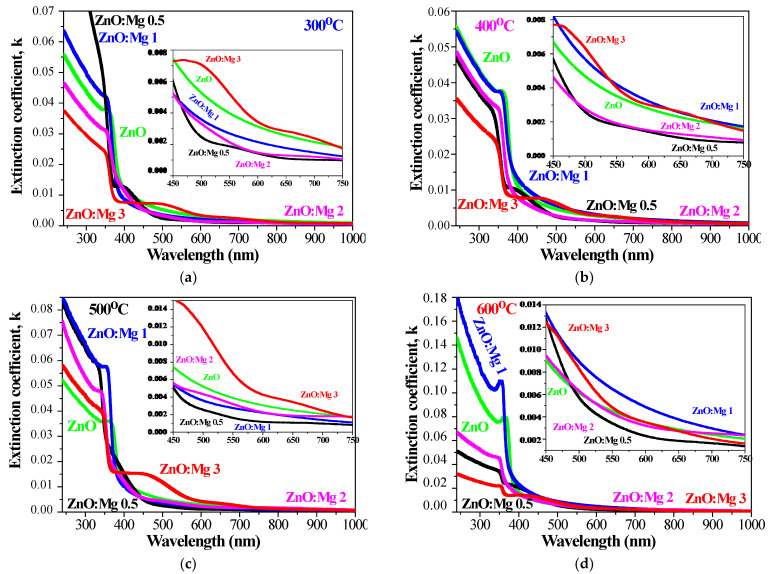
Extinction coefficient *k* of the sol–gel films of ZnO and ZnO:Mg, annealed at (**a**) 300 °C films, (**b**) 400 °C, (**c**) 500 °C and (**d**) 600 °C. The inset figures show the extinction values in the spectral range 450–750 nm.

**Figure 14 materials-15-08883-f014:**
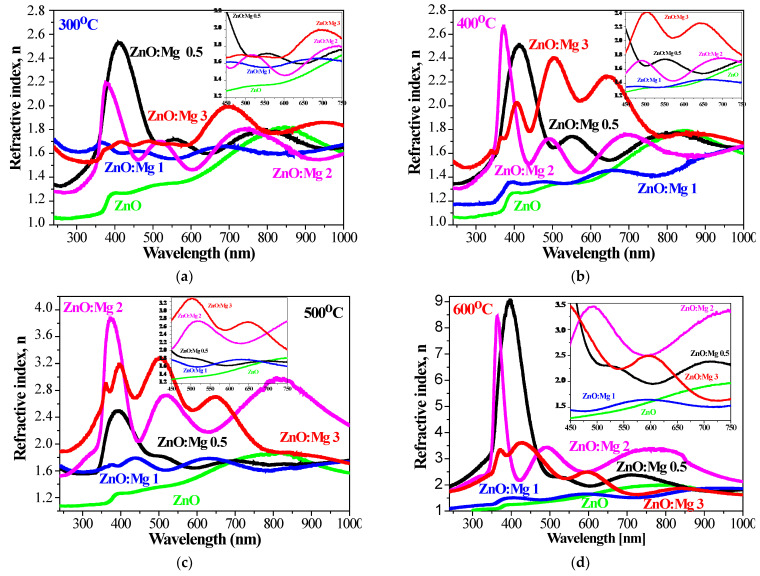
Refractive index *n* of the sol–gel films of ZnO and ZnO:Mg, treated at (**a**) 300 °C, (**b**) 400 °C, (**c**) 500 °C and (**d**) 600 °C. The inset figures show *n* values in the spectral range 450–750 nm.

**Figure 15 materials-15-08883-f015:**
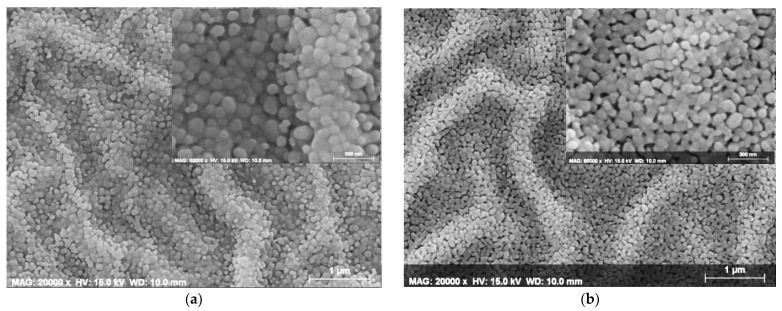
FESEM micrographs of ZnO films, deposited at a spin coating speed of (**a**) 2000 and (**b**) 4000 rpm and treated at 600 °C. The magnification is 20,000. The inset images present the film surfaces at a greater magnification of 80,000.

**Figure 16 materials-15-08883-f016:**
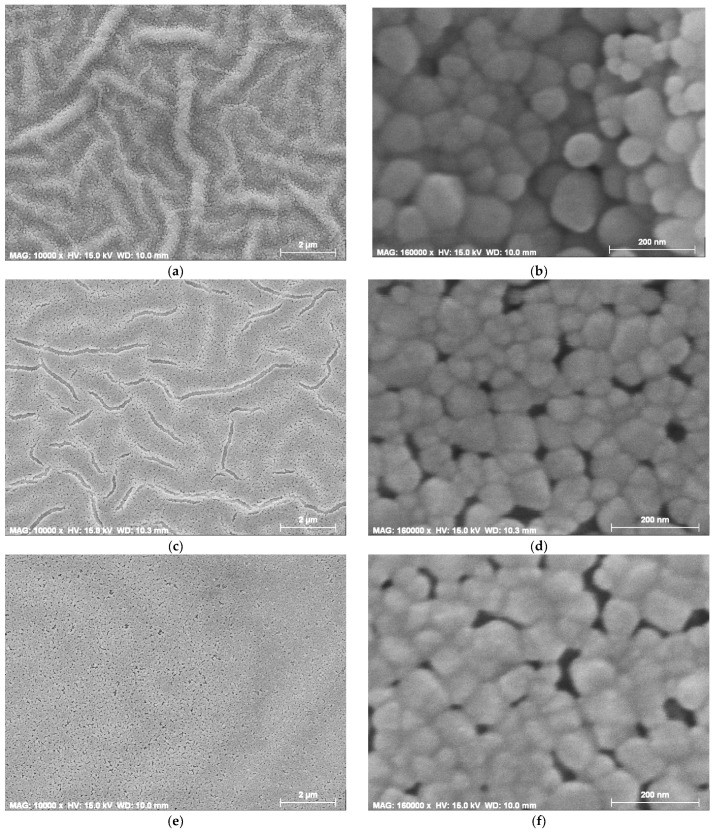
FESEM micrographs of (**a**) ZnO films, magnification 10,000; (**b**) ZnO films, magnification 160,000; (**c**) ZnO:Mg 0.5, magnification 10,000; (**d**) ZnO:Mg 0.5 films, magnification 160,000; (**e**) ZnO:Mg 3, magnification 10,000; (**f**) ZnO:Mg 3 films, magnification 160,000. The films were deposited on Si and annealed at 600 °C.

**Table 1 materials-15-08883-t001:** Comparison of the refractive index *n* of ZnO-based materials, deposited by different techniques.

Material	Deposition Method	Spectral Range (nm)	*n*	Reference
ZnO	Sol–gel spin coating	450–700	1.28–1.90	This work
ZnO:Mg			4.31–1.62	
Bulk ZnO		500	2.047–2.063	[78]
ZnO	Various	500	1.67–2.20	[78]
ZnO	Sol–gel spin coating	350–700	1.55–1.93	[82]
Zno:MnO	Sputtering	visible	2.40	[79]
ZnO:Mn	Chemical spray technique	600–1800	2.2–2.6	[83]
ZnO:Mn	Rapid thermal evaporation	600	2.07–2.148	[84]
ZnO	Ultrasonic spray pyrolysis	598	1.779	[85]
Zn1–xMg xO x = 0.10–0.16		595	1.647–1.885	[85]
ZnO:Mg, Mg 5–10%	RF magnetron sputtering	300–600	4.5–2.5	[86]
ZnO, ZnO:Na	Sol–gel spin coating	450–700	2.4–3.6	[87]
ZnO	Sol–gel, nanofibers	400–800	1.65–1.90	[88]
ZnO	Molecular beam epitaxy	300–1000	1.10–1.71	[89]
ZnO	e-beam	450–1100	2.7–2.2	[90]
(ZnO)_1−*x*_(MgO)*_x_* powder	Solid state sintering	300–1500	4.7–2.0	[91]
Zn_1-x_Cu_x_O	E-beam evaporation	450–1000	2.7–2.2	[78]
ZnO-MgO:Al_2_O_3_	RF magnetron sputtering	400–600	2.91–2.59	[6]

**Table 2 materials-15-08883-t002:** Sheet resistance, R_sheet_, average transmittance (in the spectral range 450–750 nm), T_average_, extracting the substrate transmittance, and figure of merit, FOM, determined for sol–gel ZnO and ZnO:Mg films.

	Annealing at 500 °C			Annealing at 600 °C		
Material	T_average_ (%)	R_sheet_ (Ω/sq)	FOM × 10^−4^ (Ω^−1^)	T_average_ (%)	R_sheet_ (Ω/sq)	FOM × 10^−4^ (Ω^−1^)
ZnO	74.59	685	0.78	70.48	688	0.44
ZnO:Mg 0.5	83.62	350	4.78	62.75	520	0.18
ZnO:Mg 1	78.97	694	1.36	75.10	730	0.78
ZnO:Mg 2	89.14	370	8.56	71.05	240	1.37
ZnO:Mg 3	60.11	689	0.089	66.22	680	0.24

**Table 3 materials-15-08883-t003:** Comparison of the reported values of T_average_, R_sheet_ and FOM of ZnO-based thin films.

Material	Fabrication Method	T_average_ (%)	R_sheet_ (Ω/sq)	FOM (Ω^−1^)	Reference
ZnO:Mg, F	Sputtering	90.7		1.36 × 10^−2^	[96]
ZnO	Spray pyrolysis	94.4	117	4.9 × 10^−3^	[97]
ZnO:Al	Sol–gel	82.0	156	90.3 × 10^−3^	[98]
ZnO	Spray pyrolysis	96.3	388	1.76 × 10^−3^	[99]
ZnO:Mg	Sol–gel, spin coating	97.0	1725	4.75 × 10^−4^	[100]

## Data Availability

The data are not publicly available.

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
