# Peer review of "Deposition of Sol–Gel ZnO:Mg Films and Investigation of Their Structural and Optical Properties"

_materials, 2022, doi:10.3390/ma15248883_

Round 1
Reviewer 1 Report
The authors investigated structural, vibrational, morphological, optical and electrical properties of Mg doped ZnO films prepared by sol-gel spin coating method.
This work has some merit; However it should be improved thoroughly by addressing correctly the reviewer comments:
1/ The introduction should be improved:
- The authors should introduce the storage energy applications with the others devices (Enhancing the electrical and dielectric properties of ZnO nanoparticles through Fe doping for electric storage applications, Journal of Materials Science: Materials in Electronics, 2021, 32(2), pp. 1536–1556. DOI: 10.1007/s10854-020-04923-1)
- For the improvement of optical structural and electrical properties for optoelectronic applications, the authors are recommended to add other papers with codoping like Fe/Al (Improving the optical, electrical and dielectric characteristics of ZnO nanoparticles through (Fe+Al) addition for optoelectronic applications. Applied Physics A, 2022, 128, 691, DOI: 10.1007/s00339-022-05847-9)
- Mg doped ZnO is studied in the literature so the authors should focus on the novelty of this work.
2/ The authors should take into account the strain effect for the calculus of crystallite size. They need to use Williamson Hall method to have the real size.
3/ In the majority of figures the authors should change the comma by point!
4/ The authors should add all the used equation to calculate different parameters in the manuscript.
5/ It is recommended to add cross section SEM images to deduce the thickness of each sample even at the optimum temperature and can be confirmed by optical measurement.
6/ Optical measurement that the thicknesses of the samples are not the same; is this affect the stress component of the layer?
7/ Figure 10 is not clear. We don’t need to show the bare substrate and the name of the samples should be put together and far to the curves. The authors need to zoom the ultra violet and visible zone to show the gap energy or excitonic absorption because for Mg doped ZnO the infra-red region is not necessary so the x axis should be limited to 300-1000 nm. It is recommended to put arrows for each y axis.
8/ It is mandatory to explain how the authors deduced the energy gap.
9/ In line 438: the authors claimed “The enhanced band gap can be induced by several factors:” why the authors suppose the increase of the gap energy is an enhancement? It depends on the application.
10/ Everywhere the authors used this statement “It is also well known that the….”, they should add references.
11/ In the conclusion, the authors claimed that these materials can be used in photocatalytic applications. They should explain this for Mg doped ZnO in the introduction to improve it.
Reviewer 2 Report
The manuscript “Deposition of sol-gel ZnO:Mg films and investigation of their 3 structural and optical properties” deals with the spin-coating preparation of transparent and smooth ZnO:Mg films on Si or SiO2 substrates. The manuscript presents a highly detailed study of the materials with a set of analytical methods. I recommend a major due to some concerns listed below.
1. Please focus the abstract on the specific results of the study, not on the studies themselves, like “The optical transparency, reflectance and the optical band gap of ZnO and ZnO:Mg films are investigated”.
2. In my opinion, the Introduction section lacks scientific background on the previous reports on sol-gel deposited ZnO:Mg, e.g. 10.1088/1742-6596/1428/1/012026, 10.1016/j.jallcom.2018.09.294, 10.1140/epjp/s13360-021-02208-y… Some of these references might be relevant*. Please, provide a thorough survey in view of the novelty of your manuscript. Please, make also the goal-setting of the work more clear.
*Note, I never insist that all these references are almost relevant and necessary. They just illustrate that the method was previously reported.
3. Zn sol solution – dies it contain metallic Zn? Please specify.
4. I think, more details are needed for the substrate characteristics (Si and SiO2). Maybe AFM?
5. How were the lattice parameters calculated? I hope, a full-profile refinement was used. Otherwise the results obtained by analyzing the diffraction patterns are incorrect. Please provide the experimental results along with errors. The error values are almost missing in the XRD results.
6. By the way, what was the films thickness? Why the diffraction of the substrates is absent in XRDs?
7. Was the actual chemical composition of the films studied (Zn:Mg, mol:mol)? Please also comment on the films homogeneity. Maybe EDX mapping should be applied.
8. The text “The images reveal … spin-coated ZnO films [82].” is duplicated.
9. The last sentence in Conclusions has nothing in common with the research results. Please make the conclusion section more focused, do not just list the experimental findings.
10. Concerning the manuscript logic... In my oppinion, SEM section is not in the proper place...
11. By the way, can submicron "ripples" on the surface affect optical characteristics of the materials. Maybe, this question is worth discussing...
Reviewer 3 Report
1. The reference is not well covered. The recent literature on ZnO for advanced optoelectronic applications needed to be cited. Such as https://doi.org/10.1021/acs.nanolett.9b04586
2. In figure 2c, why the ZnO:Mg3 shows the biggest peak position at the temperature of 400C? Is this data normal? Do the authors double check the data? Can the authors explain the reasons?
3. In figure 3, it seems that it has a drop in average crystalline size in 0,5% of Mg concentration and then increase in size when increasing the Mg concentration. Do the authors have any explanation on this?
4. In figure 11, the authors show the average transmittance and the reflectance, however, they are lack of the standard deviations and error bars.
5. In figure 15, the magnification area should be highlighted.
Round 2
Reviewer 1 Report
The authors corrected the manuscript following the reviewer comments. This work can be published in Materials.
Author Response
Dear Reviewer,
Please, find enclosed the revised version of the manuscript materials-2069055, entitled “Deposition of sol-gel ZnO:Mg films and investigation of their structural and optical properties” by authors Tatyana Ivanova, Antoaneta Harizanova, Tatyana Koutzarova, Benedicte Vertruyen and Raphael Closset.
We thank reviewer for detailed reviews.
Aswers to Reviewer 1
The authors corrected the manuscript following the reviewer comments. This work can be published in Materials.
We thank the reviewer for recommendations.
Sincerely yours,
Dr Tatyana Ivanova
Central Laboratory of Solar Energy and New Energy Sources,
Bulgarian Academy of Sciences,
72 Tzarigradsko Chaussee Blvd, 1784 Sofia, Bulgaria
Reviewer 2 Report
Some of my concerns were met whether some of them were not. I understand that some additional experiments requested by me can be unavailable due to the lack in time or resources... Nevertheless, I kindly ask to maje further revision in view of some unaddresseв questions.
1. Again, in the Abstract, do not mention just studies without the specific results.
2. Again, in the Introduction express directly the novelty of the work.
3. There is NO metallic (I meant elemenal) Zn in the sols as claimed. In this regard all the mentions of "Zn sol" are incorrect.
4. At least the source of Si and SiO2 wafers must be provided.
5. Ref. 41 is confusing - did Shkir et al. proposed Scherrer eqn.? I am still confused on the use of the least square method for the assessment of crystal parameters. Even no tetails on the deconvolution method is given in the experimental section. Full profile refinement is still needed.
6. How exactly the film thickness was evaluated?
Author Response
Please, see the attachment
